# Lifetime Weight Cycling and Central Fat Distribution in Females With Obesity: A Brief Report

**DOI:** 10.3390/diseases8020008

**Published:** 2020-03-26

**Authors:** Hana Tannir, Leila Itani, Dana El Masri, Dima Kreidieh, Marwan El Ghoch

**Affiliations:** Department of Nutrition and Dietetics, Faculty of Health Sciences, Beirut Arab University, Beirut P.O. Box, 11-5020 Riad El Solh, Lebanon; hana.tannir@bau.edu.lb (H.T.); l.itani@bau.edu.lb (L.I.); dana.masri@bau.edu.lb (D.E.M.); d.kraydeyeh@bau.edu.lb (D.K.)

**Keywords:** body composition, BMI, obesity, weight cycling, body fat distribution

## Abstract

Weight cycling (WC) is a common phenomenon in patients with obesity, however, its consequence on body composition has not yet been fully understood. Therefore, we aimed to determine whether multiple WC can negatively affect the latter, especially in terms of body fat distribution in female adults seeking treatment that are overweight or obese. Body composition was obtained using a segmental body composition analyser (MC-780MA, Tanita Corp., Tokyo, Japan) in 125 adult females who had been referred to the Department of Nutrition and Dietetics at the Beirut Arab University (Lebanon). WC was defined as intentional weight loss of ≥3 kg followed by involuntary weight regain of ≥3 kg, and participants were categorized as WC if they had experienced ≥2 cycles. Ninety of the 125 participants met the criteria for WC and displayed a higher total and trunk fat mass than those without WC. This was confirmed through linear regression analysis, showing that multiple WC were associated with increased fat mass (FM) by nearly 4.2 kg (β = 4.23, 95%CI: 0.81–7.65, *p* = 0.016)–2.4 kg in the trunk region (β = 2.35, 95%CI: 0.786–3.917, *p* = 0.004) when compared to the non-WC group, after adjusting for age and fat-free mass. In conclusion, multiple WC is associated with increased body fat, especially in the central region. Future studies are needed to examine the impact of this fat distribution on health outcomes in this phenotype of patients.

## 1. Introduction

Obesity is defined as increased fat mass in the adipose tissue [1] and it is becoming a rising health problem worldwide [2]. Global prevalence has nearly doubled in more than 70 countries and its prevalence is continuously increasing in many others [3], to the extent that it has tripled in some developing and middle-income countries since the early 1980s [3]. Obesity is associated with a high incidence of medical [4] and psychosocial comorbidities [5] which seem to increase the rates of mortality [6]. Therefore, the World Health Organization (WHO) considers the priority of addressing this problem as urgent, and states that this should be done through strong action in order for it to be translated through meaningful changes by individuals, health-care task forces, and policymakers.

This has led to international guidelines recommending a wide range of weight-loss interventions. These include lifestyle modification programs, which is mainly considered the key element of weight management for patients with obesity [7], as well as anti-obesity drugs [8]. and bariatric surgery [9]. However, it is widely known that although most weight-loss approaches are effective in the short term, weight regain is a common phenomenon in patients who have intentionally lost weight, with most patients returning to their baseline weight within 3–5 years of follow up [10].

This repeated failure in the long-term maintenance of weight loss leads to body weight fluctuation over time, termed ‘weight cycling’ (WC) [11]. In the last few decades, the study of this phenomenon has gained particular attention, and several pieces of research have attempted to clarify whether WC may adversely affect health outcomes [12,13,14,15,16]. However, such research has produced inconsistent evidence, especially in terms of changes in body composition patterns. In particular, body fat distribution is still unclear.

In light of these considerations, the current study aims to investigate the impact of lifetime WC on body composition, specifically body fat and body fat distribution in the ‘real-world’ clinical setting of treatment-seeking female patients that are overweight or obese.

## 2. Materials and Methods

A total of 125 female patients that were overweight or obese and had sought weight-loss treatment, were recruited consecutively after being referred by general practitioners to the Nutritional and Weight Management Outpatient Clinic in the Department of Nutrition and Dietetics at the Beirut Arab University (BAU) in Lebanon, during the period of May 2017 to January 2020. The patients who were eligible for this study were females, aged ≥18 years, with a BMI ≥25.0 kg/m^2^, and with at least one weight-loss responsive comorbidity (i.e., type 2 diabetes, cardiovascular disease, sleep apnoea, severe joint disease, or two or more risk factors), as defined by the Adult Treatment Panel III [17]. Patients were excluded if they were male, pregnant or lactating, taking medication that affects body weight, or if they presented with medical co-morbidities associated with weight loss, or severe psychiatric disorders. The study was approved by the Institutional Review Board of the Beirut Arab University (no. 2017H-0034-HS-R-0241), and all participants gave informed written consent.

A questionnaire was administered to participants in order to gather information regarding their social demographic (age, marital status, employment, level of education, etc.). Moreover, their prior intentional weight-loss attempts were carefully evaluated by a trained dietician involved in the study, and WC was deemed to be intentional weight loss of ≥3 kg, followed by involuntary weight regain of ≥3 kg [16]. Participants were considered multiple WC if they experienced ≥2 cycles [16].

Body weight and height were measured using an electronic weighing scale (SECA 2730-ASTRA, Hamburg, Germany) and a stadiometer. Then, BMI was calculated according to the standard formula of body weight measured in kilograms, divided by the square of the height in meters.

Body composition was measured using a segmental body composition analyser (MC-780MA, Tanita Corp., Tokyo, Japan) [18]. After the gender, age, and height information had been entered into the device, participants were asked to stand in a stable position in bare feet. The device provided separate body mass readings for different segments of the body, using an algorithm incorporating impedance, age, and height, to estimate the total and regional fat mass (FM) and fat-free mass (FFM) [18].

Cardiometabolic disease in this study is defined as the presence of any diseases, such as type 2 diabetes, cardiovascular diseases (coronary heart disease, stroke, transient ischemic attack, and peripheral arterial disease), and dyslipidemia (a decreased concentration of high-density lipoprotein cholesterol, and an increased concentration of high-density lipoprotein cholesterol and triglycerides), based on self-reported diagnosis, either simultaneously or separately.

### Statistical Analysis

The normality of the data was checked using Shapiro–Wilk or Kolmogorov–Smirnov tests, as well as a quantile–quantile (Q–Q) plot. Age, FM and WC frequency, minimum weight achieved, and weight at age 20 years did not satisfy the normality criteria. Frequencies, means, standard deviations, medians, and interquartile ranges were used to describe the anthropometric characteristics of the study population as appropriate. Means, medians, and frequencies were compared using a student’s t test, a Mann–Whitney U test, and a Chi squared test, respectively. Simple (Model I) and multiple linear regression analyses were used to assess the association between anthropometric characteristics including FM (kg), %FM, trunk fat (kg), and % trunk fat as dependent variables, and being WC as an independent variable, while adjusting for other covariates including age and FFM (Models I and II). Diagnostic tests of the regression assumption for linearity, equal variance, normality of residuals, and the variance inflation factor (VIF) for testing collinearity between independent variables were conducted. The ANOVA was used to test if the variance explained was different from zero (R^2^ ≠ 0). All analyses were performed using SPSS version 26.0 (IBM Corp.; IBM, Armonk, NY, USA). Tests were considered statistically significant at *p* < 0.05.

## 3. Results

The socio-demographic and anthropometric characteristics of the study population are presented in Table 1. The age did not vary between the WC group (37.12, IQR: 26.79 years) and the non-WC group (41.15, IQR: 22.77 years). The WC group had a higher level of education (41.1% vs. 18.2%) and more were unmarried (47.8% vs. 24.2%). Moreover, the WC group had a higher mean weight (90.89 ± 12.56 vs. 86.23 ± 10.35 kg), FM (38.20, IQR: 11.20 kg vs. 34.80, IQR: 10.30 kg), and trunk fat (18.03 ± 4.32 vs. 15.51 ± 3.22), and lower %FFM (56.81 ± 5.08% vs. 59.33 ± 4.18%). The mean %FM (42.76 ± 5.19% vs. 40.67 ± 4.18%) and % trunk fat (39.47 ± 6.90 vs. 35.81 ± 5.41) were also significantly higher among the WC group, compared to the non-WC group (Figure 1 and Figure 2). Finally, the WC group had a higher weight at age 20 (66.75, IQR: 24.00 vs. 60.00, IQR: 13.00) and had a higher frequency of weight cycling (5.00, IQR: 3.00 vs. 0, IQR: 1.00).

Table 2 presents the simple and multiple linear regression models, showing a significant association between WC and FM compartments. The results indicate that being a weight cycler increases FM by 4.2 kg (β = 4.23, 95%CI: 0.81–7.65, *p* = 0.016), trunk fat by 2.4 kg (β = 2.35, 95%CI: 0.786–3.917, *p* = 0.004), and % trunk fat by 3.6% (β = 3.57, 95%CI: 1.07–6.077, *p* = 0.005). The associations were significant after adjustment for different confounders, including age and FFM, in all the models except for %FM, since the model was attenuated after adjusting for age and FFM (R^2^ = 0.036, *p* = 0.212).

## 4. Discussion

Our study aimed to provide preliminary data on the association between multiple WC and body composition patterns in adult females that were overweight or obese, and one major finding was revealed.

The group of participants with multiple WC experienced higher FM, especially in the central region, compared to their non-WC counterparts. In fact, the presence of the multiple WC status increased the FM deposition particularly in the trunk region when compared with those without WC, while controlling for age and FFM. This is in line with a previous large sampled cross-sectional study [19], that found that multiple WC, defined as self-reported events of at least five any-type intentional weight loss episodes of ≥5 kg followed by rapid return to baseline or higher body weight, was associated with abdominal fat accumulation. This study relied on indirect assessment of the latter, namely by means of the waist circumference and waist-to-hip ratio assessment [19]. On the other hand, our finding contrasts those of other studies that found that weight regain that followed the period of intentional weight loss did not negatively change body composition (i.e., by increasing fat mass) [15,20,21,22,23]. However, despite the robust design of these studies, some were composed of limited sample size (i.e., 10–20 participants), or investigated “induced” short-term WC (i.e., one cycle) with intentional weight regain. This clearly cannot be representative of the natural history of obesity, in which WC occurs unintentionally in multiple ways, and over a variable period of time (i.e., over months or years) [15,20,21,22,23].

Our study has certain strengths. Principally, to the best of our knowledge, it is one of the few studies to assess the impact of multiple WC on body composition in treatment-seeking female patients that are overweight or obese, in a ‘real-world’ clinical setting. However, our study did have some limitations. First, our sample included only female patients seeking an outpatient weight management treatment program, hence, our findings are not extendable to patients with obesity of different genders (i.e., males) or those who seek other treatment modalities (i.e., bariatric surgery, pharmacological interventions, etc.). Second, the information relative to WC was self-reported and did not rely on objective assessment. Third, we relied on an arbitrary definition of WC, which is a common limitation of the studies on this topic due to the lack of a standardized definition of WC. Fourth, we assessed body composition using the impendence analyser. Despite its validation vs. dual-energy X-ray absorptiometry (DXA) scan [18], this is still not accepted as a gold-standard technique for patients with obesity. Fifth, the cross-sectional design of our study should be considered as another limitation. Sixth, no biochemical testing (i.e., hormonal levels) was conducted to exclude secondary obesity. Moreover, we are unable to shed any light on the mechanisms behind the differences in body fat distribution that we observed in weight cycling adult females that were overweight or obese. Finally, in the diagnosis of cardiometabolic diseases, we relied on self-reporting data.

## 5. Conclusions

In our study, we provide evidence that a lifetime of multiple WC may lead to unfavorable body composition patterns, namely central fat distribution in treatment-seeking females that are overweight or obese. Future studies are needed to clarify whether this particular type of body fat accumulation can influence the development of metabolic and cardiovascular complications.

## Figures and Tables

**Figure 1 diseases-08-00008-f001:**
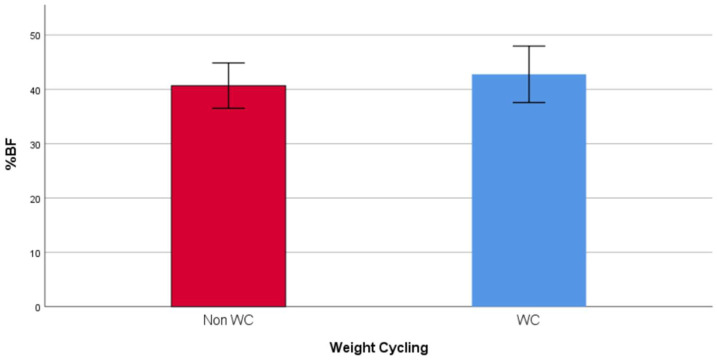
Mean %FM in WC and non-WC groups.

**Figure 2 diseases-08-00008-f002:**
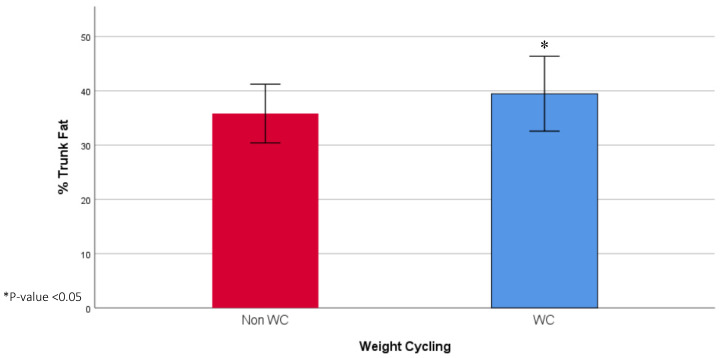
Mean % trunk fat in WC and non-WC groups.

**Table 1 diseases-08-00008-t001:** Socio-demographic, anthropometric, and body composition characteristics of the study population (*N* = 125) *^§.^

	Total (N = 125)	WC (N = 90)	Non-WC (N = 35)	Significance
Socio-demographic				
Age ^§^	38 (27.34)	37.12 (26.79)	41.15 (22.77)	P = 0.301
Level of education				X^2^ = 5.58; P = 0.018
Lower Education	80 (65.0)	53 (58.9)	27 (81.8)	
University	43 (35.0)	37 (41.1)	6 (18.2)	
Marital status				X^2^ = 5.51; P = 0.019
Unmarried	51 (41.5)	43 (47.8)	8 (24.2)	
Married	72 (58.5)	47 (52.2)	25 (75.8)	
Employment				X^2^ = 0.031; P = 0.860
Unemployed	88 (71.5)	64 (71.1)	24 (72.7)	
Employed	35 (28.5)	26 (28.9)	9 (27.3)	
Body composition				
Weight (kg)	89.58 (12.13)	90.89 (12.56)	86.23 (10.35)	0.037
BMI (kg/m^2^)	35.73 (4.66)	35.79 (4.56)	35.58 (4.99)	0.827
FM (kg) ^§^	37.60 (11.80)	38.20 (11.20)	34.80 (10.30)	0.02
FM (%)	42.18 (5.00)	42.76 (5.19)	40.67 (4.18)	0.022
FFM (kg)	51.17 (5.57)	51.26 (5.65)	50.95 (5.43)	0.782
FFM (%)	57.52 (4.96)	56.81 (5.08)	59.33 (4.18)	0.006
Trunk Fat	17.32 (4.18)	18.03 (4.32)	15.51 (3.22)	0.001
% Trunk Fat	38.44 (6.70)	39.47 (6.90)	35.81 (5.41)	0.002
Body weight history				
Minimum weight achieved (kg) ^§^	70.00 (16.00)	70.00 (13.5)	70.00 (20.00)	0.850
Maximum weight achieved (kg)	92.08 (12.53)	93.35 (12.14)	88.51 (12.64)	0.067
Weight at age 20 yrs. (kg) ^§^	64.00 (21.00)	66.75 (24.00)	60.00 (13.00)	0.002
Weight cycling frequency ^§^	3.00 (5.00)	5.00 (4.00)	0 (1.00)	<0.0001
Cardiometabolic disease				X^2^ = 0.096; P = 0.757
No	109 (87.2)	79 (87.8)	30 (85.7)	
Yes	16 (12.8)	11 (12.2)	5 (14.3)	

* Values are N (%) for categorical variables and Mean (SD) or ^§^ Median (IQR) for continuous variables. ^§^ P values are results for Mann Whitney test. BMI = body mass index, FM = fat mass, FFM = fat free mass, and WC = weight cycling.

**Table 2 diseases-08-00008-t002:** Linear regression coefficients for the association between different fat mass (FM) compartment and weight cycling (WC).

	Model I			Model II			Model III		
	Beta	95%CI	P	Beta	95%CI	P	Beta	95%CI	P
FM									
WC	4.489	0.832, 8.146	0.017	4.229	0.566, 7.891	0.024	4.229	0.805, 7.654	0.016
Age				−0.074	−0.180, 0.031	0.166	−0.021	−0.122, 0.081	0.690
FFM							0.619	0.334, 0.903	0.000
% FM									
WC	2.087	0.141, 4.032	0.036	2.120	0.157, 4.083	0.034	2.120	0.149, 4.091	0.035
Age				0.010	−0.047, 0.066	0.739	0.009	−0.050, 0.067	0.773
FFM							−0.011	−0.175, 0.152	0.890
Trunk Fat									
WC	2.519	0.925, 4.114	0.002	2.351	0.770, 3.933	0.004	2.351	0.786, 3.917	0.004
Age				−0.048	−0.094, −0.002	0.039	−0.37	−0.084, 0.009	0.116
FFM							0.123	−0.007, 0.253	0.063
% Trunk Fat									
WC	3.655	1.084, 6.227	0.006	3.575	0.983, 6.166	0.007	3.574	1.072, −6.077	0.005
Age				−0.023	−0.098, 0.052	0.543	−0.052	−0.126, 0.023	0.172
FFM							−0.329	−0.537, −0.121	0.002

FM = fat mass, FFM = fat free mass, WC = weight cycling.

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
