# Peer review of "Lifetime Weight Cycling and Central Fat Distribution in Females With Obesity: A Brief Report"

_diseases, 2020, doi:10.3390/diseases8020008_

Round 1
Reviewer 1 Report
Reviewer’s report
Title:
Lifetime Weight Cycling and Central Fat Distribution
in Females with Obesity: a Brief Report
Authors:
Hana Tannir, Leila Itani, Dana El Masri, Dima Kreidieh, Marwan El Ghoch
Weight cycling accompanies weight loss attempts, and epidemiological data suggest that it can be more detrimental for the whole body function that remaining overweight or obese. The pathomechanisms underlying this phenomenon remain largely unknown. In their study, Tannir et al. found that weight cycling unfavorable influence body composition, predisposing to the accumulation of visceral fat. This finding constitutes a contribution to our understanding of mechanisms responsible for the adverse health effects of weight cycling. However, before publication, some important issues should be considered.
Major remarks:
Section: Material and Methods
1) Please provide more detailed characteristics of the study participants:
- if all of them were diagnosed with simple obesity? Did they undergo any hormonal evaluation to exclude secondary obesity?
- what was the prevalence of obesity-associated co-morbidities in the study group?
- did the diabetic study participants take metformin? Metformin is a primary treatment for type 2 diabetic patients and may favorably influence body mass and composition.
2) Was the power calculation performed to verify if the sample size was sufficient to obtain reliable results?
Section: Results
- Please correct the data in the text and Table 1 regarding the marital status of study participants.
- Is it possible to evaluate if the unfavorable changes in body composition and adipose tissue distribution in the WC group translated into the higher prevalence of obesity-associated co-morbidities?
- Please make the abbreviations consistent: in Tables abbreviation, FM (fat mass) is used, while within the text and in Figure 1 – BF (presumably referring to body fat).
Section: Discussion
The main limitation of the study is associated with the fact that it is not longitudinal. Since the initial body composition and adipose tissue distribution of study participants is unknown, it is not possible to asses that the observed difference between the two studied groups is associated with the history of weight-loss interventions.

Author Response
Reviewer 1
Major remarks:
Section: Material and Methods
- Please provide more detailed characteristics of the study participants:
- If all of them were diagnosed with simple obesity? Did they undergo any hormonal evaluation to exclude secondary obesity?
Response: No biochemical analysis has been preformed to exclude secondary obesity. Now we report this among the limitations in the Discussion section.
- What was the prevalence of obesity-associated co-morbidities in the study group?
Response: We added in table 1 the self-reported cardiometabolic diseases obtained from patients’ medical history record.
- Did the diabetic study participants take metformin? Metformin is a primary treatment for type 2 diabetic patients and may favorably influence body mass and composition.
Response: Very few participant self-reported pre-diabetes or diabetes (5-6 participants) among the entire sample and comparison between subgroups (WC vs. Non WC) become meaningless.
- Was the power calculation performed to verify if the sample size was sufficient to obtain reliable results?
Response: The minimum sample size for linear regression analysis using 3 predictors with 80 % power and an α=0.05 was calculated using G Power software. Based on the R2 of each model, the sample sizes needed were n= 63, n= 88 and n= 86 for the FM, Trunk Fat and %Trunk Fat models respectively.
Section: Results
- Please correct the data in the text and Table 1 regarding the marital status of study participants.
Response: Done as suggested.
- Is it possible to evaluate if the unfavorable changes in body composition and adipose tissue distribution in the WC group translated into the higher prevalence of obesity-associated co-morbidities?
Response: Done as suggested. No significant difference prevalence of obesity-associated between WC vs. Non WC groups (Table 1).
- Please make the abbreviations consistent: in Tables abbreviation, FM (fat mass) is used, while within the text and in Figure 1 – BF (presumably referring to body fat).
Response: Done as suggested.
Section: Discussion
The main limitation of the study is associated with the fact that it is not longitudinal. Since the initial body composition and adipose tissue distribution of study participants is unknown, it is not possible to assess that the observed difference between the two studied groups is associated with the history of weight-loss interventions.
Response: We added the issue raised by the reviewer to the limitations.
Reviewer 2 Report
This manuscript by Tannir et al performed a study to figure out the relationship between weight cycling and change of body fat composition. Overall, this study has a good experimental design and its unique strength compared to other studies. Here are some suggestions for this manuscript.
Minor suggestions
- Line 17, 3 kg should change to 3 Kg
- Discuss the reasons why other published articles in weight cycling (Ref. 12-16) have different results in body composition. The authors did discuss Ref.15 in the Discussion. How about others?
- Line 100, mean %BF, please define BF when first use this term. But in Table 1, there is no BF. I think here body fat (BF) should be the same as fat mass (FM). Please be consistent throughout the manuscript.
- What is the difference between Model I to III in Table 2. This should be addressed in Methods.
- The authors did not mention Figure 1 and 2 in the main text. And the results in Figure 1 and 2 actually came from Table 1. These figures are redundant
Author Response
Reviewer 2
Minor suggestions
Line 17, 3 kg should change to 3 Kg
Response: Done as suggested.
Discuss the reasons why other published articles in weight cycling (Ref. 12-16) have different results in body composition. The authors did discuss Ref. 15 in the Discussion. How about others?
Response: The other published articles that have different results on body composition (15, 20-23), has been discussed.
Line 100, mean %BF, please define BF when first use this term. But in Table 1, there is no BF. I think here body fat (BF) should be the same as fat mass (FM). Please be consistent throughout the manuscript.
Response: Done as suggested.
What is the difference between Model I to III in Table 2. This should be addressed in Methods.
Response: Now we added as suggested.
The authors did not mention Figure 1 and 2 in the main text. And the results in Figure 1 and 2 actually came from Table 1. These figures are redundant
Response: Now figures 1 and 2 are mentioned in the main text. We agree with reviewer that the results showed in the figures are in Table 1 too, however these put emphasis on the only two variables that associated with WC. For this reason we kindly ask to keep them in the final version.
Round 2
Reviewer 1 Report
The Authors took into account my previous comments regarding the manuscript, and therefore, in my opinion, it meets the criteria necessary for publication in the Diseases Journal.